# Beyond Bullying, Aggression, Discrimination, and Social Safety: Development of an Integrated Negative Work Behavior Questionnaire (INWBQ)

**DOI:** 10.3390/ijerph20166564

**Published:** 2023-08-11

**Authors:** Cokkie Verschuren, Maria Tims, Annet H. De Lange

**Affiliations:** 1Department of Management and Organization, School of Business and Economics, Vrije Universiteit Amsterdam, 1081 HV Amsterdam, The Netherlands; 2The Faculty of Psychology, Open University, 6419 AT Heerlen, The Netherlands; 3The Department of Psychology, Universidade da Coruna, 15701 A Coruña, Spain; 4The Faculty of Psychology, Norwegian University of Science and Technology (NTNU), 7491 Trondheim, Norway; 5Norwegian School of Hotel Management, University of Stavanger, 4021 Stavanger, Norway; 6Department of Human Resource Studies, Faculty of Social and Behavioral Sciences, Tilburg University, 5037 AB Tilburg, The Netherlands

**Keywords:** diagnostic instrument, employee wellbeing, harm, negative work behavior (NWB), bystander roles

## Abstract

Negative work behavior (NWB) threatens employee well-being. There are numerous constructs that reflect NWBs, such as bullying, aggression, and discrimination, and they are often examined in isolation from each other, limiting scientific integration of these studies. We aim to contribute to this research field by developing a diagnostic tool with content validity on the full spectrum of NWBs. First, we provide a full description of how we tapped and organized content from 44 existing NWB measurement instruments and 48 studies. Second, we discussed our results with three experts in this research field to check for missing studies and to discuss our integration results. This two-stage process yielded a questionnaire measuring physical, material, psychological, sociocultural, and digital NWB. Furthermore, the questions include a range of potential actors of NWB, namely, internal (employees, managers) and external actors (clients, customers, public, and family members) at work and their roles (i.e., target, perpetrator, perpetrator’s assistant, target’s defender, outsider, and witness of NWBs). Finally, the questionnaire measures what type of harm is experienced (i.e., bodily, material, mental, and social harm).

## 1. Introduction

A broad range of negative work behaviors (NWBs) are currently being studied by clinicians and researchers, such as victimization [1], bullying [2], aggression [3], and deviance [4]. Although all these separate studies provide important knowledge about the prevalence and consequences of these negative work behaviors, the limitations of this narrow approach become irrefutably clear.

First, researchers have indicated that various types of NWB occur simultaneously and sequentially [5,6,7,8,9]. For instance, swearing (psychological verbal) and hitting (physical attack) often occur simultaneously. Another example of a combination of NWBs is found in #MeToo, where traditional types of negative (work) behaviors, such as physical (rape), material (no promotion), and psychological (gossiping) NWBs, alternate with digital NWBs, such as pornographic pictures via GSM (global system for mobile communications) [10]. This shows that the real and online worlds also merge or occur simultaneously and sequentially [11]. Further supporting the co-occurrence of NWBs, Privitera and Campbell [12] found that 5.8% of their respondents reported experiencing only one type of negative act, while 83.5% reported experiencing two or more types of real/online negative acts.

Second, next to the often isolated focus on negative work behaviors instead of their co-occurrence, existing measurement instruments usually focus on one actor, such as organizational actors (e.g., Negative Acts Questionnaire: NAQ-R) [13] or clients (e.g., Healthcare-worker’s Aggressive Behaviour Scale-Users) [14], whereas multiple actors are often involved in NWBs. These actors vary between strangers/public, workers/managers, clients/pupils/customers, and relatives of those as an individual or as a group [15].

A disadvantage of measuring actor types separately is that it ignores the increasingly blurring boundaries between internal and external actor types in modern workplaces. Examples of these workplaces are the healthcare [16], the police [17], education [18], or the public sector in general [19]. In these workplaces, external actors were found to play a key role in employees’ interactions with each other, with customers, and with actor relatives [20]. Therefore, the dynamics with external actor types, such as the public, clients, students, customers, providers, and their relatives, should be included in research on NWBs to have a complete view of NWBs.

A third limitation of the focus on measuring separate NWBs is that limited insights are gained on the specific nature of their inflicted harm on different actors and their roles. Previous research has shown that NWB inflicts harm on targets [21], perpetrators [22], bystanders as witness [23], companies [24], and society [25]. However, the current questionnaires often limit their focus to the harm of targets, thus only collecting data on one source [26]. Another aspect is that scientists have indicated that multiple types of harm, such as material (e.g., job loss) and psychological harm (e.g., PTSD), occur in combination [27,28]. From these insights, it is important to measure harm among various actors and their roles.

As a final limitation, as these constructs of NWB overlap, separate measurements offer no discriminating validity. This may create the problem that bullying or sexual harassment also have features of interpersonal conflict or aggression, which are considered different types of NWB. Based on the above observed challenges in measuring NWBs, and to further scientific knowledge on this important topic, the purpose of this study is to develop a full-spectrum diagnostic instrument to assess NWBs. This developmental breadth was searched for in the coverage of NWB types with co-occurrence patterns, types of harm, actor types, and actor roles in accordance with the model of Verschuren et al. [29]. These authors reviewed the NWB field and identified the overlapping and unique aspects of the operationalizations of NWBs to specify a new integrative definition of NWB. The key elements in their definition are that NWBs are (1) distinguished in five natures (i.e., physical, material, psychological, sociocultural, and/or digital); (2) are associated with different types of harm (i.e., physical, psychological, material, and social); (3) are engaged in by different actors (i.e., criminal/stranger, customer/client/pupil, coworker/manager, or relative); (4) can be performed by individuals, dyads, triads, or groups; and (5) are often taking place over time.

Thus, using this model as the guiding framework of this study, fully examining the concept of NWB requires measurement of the combination of the different NWB types; they can be physical (e.g., kicking), material (e.g., littering), psychological (e.g., isolating), sociocultural (e.g., discriminating on gender), or digital (e.g., hacking), each occurring in different patterns. These patterns of negative work behaviors tend to be systematic (e.g., not a one-time event but a repeated process [30]); tend to be ongoing (e.g., occurrence of incidents during a year) [31,32]; tend to escalate into more serious NWB (e.g., worsens from verbal into physical acts) [33]; and vary from being overtly visible, such as publicly criticizing [34], to covert and less visible forms as a silent treatment [35].

Furthermore, a full measure of NWB should include the different types of harm, namely, physical (e.g., cardiovascular disease) [36], material (e.g., replacement costs by employee turnover) [37], psychological (e.g., post-traumatic stress disorder) [38], and social harm (e.g., family consequences) [39]. Additionally, the NWB measure should include the measurement of different actor types involved in NWBs, i.e., strangers/public (e.g., visitor) [14], workers/managers (e.g., teacher) [40], clients/pupils/customers (e.g., supplier) [41], and relatives (e.g., friend) [14]. Finally, the actor roles need to be assessed, which requires the measurement of the roles of the target, perpetrator, and bystander as witness, assistant, defender, and outsider. This means that a respondent to the NWB questionnaire can have different roles (e.g., target/perpetrator).

To achieve our goal of developing a full-spectrum diagnostic instrument to assess NWBs in an integrative way, we thoroughly screened and analyzed existing instruments and studies on digital NWB and its inflicted harm to build subcategories and select measurement items that fit an integrated NWB questionnaire. We also report adjustments that were made to existing items or creation of new items. To validate our efforts, we next report the results of an expert panel who thoroughly evaluated the development of the integrated NWB questionnaire (INWBQ).

This article is organized as follows. First, we explain our development method for the INWBQ on dimension and item selection. Then, we present our results for the dimensions of NWB, occurrence patterns, harm, actor types, and actor roles. Next, we explain which adjustments were made and why. Thereafter, we indicate how we processed the contributions from the expert panel. We end with a discussion about scientific and practical implications, limitations, and future research.

## 2. Method

To derive an item pool for an integrated NWB instrument, as indicated above, we used the definition of Verschuren et al. [29] as a starting point. Using these aspects of NWBs, we examined the literature to identify measurement instruments that can serve as the basis for our INWBQ. Below, we outline the different steps we took to identify these instruments.

### 2.1. Selection of Instruments

Inclusion of construct instruments and studies:

We looked for questionnaires to fill all aspects from the definition of NWB, i.e., type of NWB, harm of NWB, types of actors, actor roles, and occurrence patterns. We searched for these instruments in the review of Verschuren et al. [29] that identified the 16 most frequently cited NWB constructs as keywords [29]:(1)Workplace aggression, i.e., “efforts by individuals to harm others with whom they work, or have worked, or the organizations in which they are currently, or were previously, employed. This harm-doing is intentional and includes psychological as well as physical injury” [42] (p. 38).(2)Bullying, i.e., “harassing, offending, socially excluding someone or negatively affecting someone’s work. It has to occur repeatedly and regularly (e.g., weekly) and over a period of time (e.g., about six months). Bullying is an escalating process in the course of which the person confronted ends up in an inferior position becoming the target of systematic negative social acts. A conflict cannot be called bullying if the incident is an isolated event or if two parties of approximately equal ‘strength’ are in conflict” [43] (p. 26).(3)Mobbing, i.e., “situations where a worker, supervisor, or manager is systematically and repeatedly mistreated and victimized by fellow workers, subordinates, or superiors. The term is widely used in situations where repeated aggressive and even violent behavior is directed against an individual over some period of time” [44] (p. 379).(4)Harassment/discrimination, i.e., “interpersonal behavior aimed at intentionally harming another employee in the workplace” [45] (p. 998).(5)Workplace deviance, i.e., “voluntary behavior that violates significant organizational norms and, in so doing, threatens the well-being of the organization or its members, or both” [46] (p. 555).(6)Counterproductive work behavior, i.e., “volitional acts that harm or are intended to harm organizations or people in organizations” [47] (p. 151).(7)Workplace violence, i.e., “the act or threat of violence, ranging from verbal abuse to physical assaults directed toward persons at work or on duty. The impact of workplace violence can range from psychological issues to physical injury, or even death” [48] (p. 1).(8)Abuse, i.e., “interactions between organizational members that are repeated hostile verbal and nonverbal, often nonphysical behaviors directed at a person(s) such that the target’s sense of him/herself as a competent worker and person is negatively affected” [49] (p. 212).(9)Terror, i.e., “hostile and unethical communication which is directed in a systematic way by one or a number of persons mainly toward one individual. These actions take place often (almost every day) and over a long period (at least for six months) and, because of this frequency and duration, result in considerable psychic, psychosomatic and social misery” [50] (p. 120).(10)Injustice, i.e., violating distributive, procedural, and interpersonal rules [51]. Distributive injustice, i.e., “ dissatisfaction and low morale related to a person’s suffering injustice in social exchanges… felt injustice is a response to a discrepancy between what is perceived to be and what is perceived should be (e.g., effort and reward, allocation of scarce rewards between different people)” [52] (p. 272); Procedural justice, i.e., “members’ sense of the moral propriety of how they are treated—is the “glue” that allows people to work together effectively. Justice defines the very essence of individuals’ relationship to employers” [53] (p. 34). In contrast, injustice (e.g., inconsistent treatment, discrimination or ill-treatment, imprecision, ethical flaw, or prejudice), is like “a corrosive solvent that can dissolve bonds within the community. Injustice is hurtful to individuals and harmful to organizations” [53] (p. 36); Interpersonal justice, i.e., “treating an employee with dignity, courtesy, and respect”. This includes “informational justice: sharing relevant information with employees.” [53] (p. 36).(11)Interpersonal conflict, i.e., “range from minor disagreements between coworkers to physical assaults on others. The conflict may be overt (e.g., being rude to a coworker) or may be covert (e.g., spreading rumors about a coworker)” [54] (p. 357).(12)Victimization/scapegoating,-victimization i.e., “an employee’s perception of having been the target, either momentarily or over time, of emotionally, psychologically, or physically injurious actions by another organizational member with whom the target has an ongoing relationship” [55] (p. 1023); -scapegoating: i.e., “an extreme form of prejudice in which an outgroup is unfairly blamed for having intentionally caused an ingroup’s misfortunes” [56] (p. 244).(13)Micropolitics, i.e.,” referring to employees’ perceptions in organizations, they often describe political behaviors in negative terms and associate these with self-serving behaviors, usually at the expense of others” [57] (p. 139).(14)Ostracism i.e., “the exclusion, rejection, or ignoring of an individual (or group) by another individual (or group) that hinders one’s ability to establish or maintain positive interpersonal relationships, work-related success, or favorable reputation within one’s place of work” [58] (p. 217).(15)Incivility, i.e., “low-intensity deviant behavior with ambiguous intent to harm the target, in violation of workplace norms for mutual respect. Uncivil behaviors are characteristically rude and discourteous, displaying a lack of regard for others” [59] (p. 457).(16)Social safety, i.e., “in a systemic or socio-ecological approach [60,61]), the dyad (victim and perpetrator), the triad (dyad and bystander) and the group are placed in larger systems around them such as the school, neighborhood and society. Therefore, power relations are influenced by the cultural and social context” [62] (p. 109), [63].

We focused on the existing published measurement instruments of these constructs, preferably validated ones. This search yielded a total of 34 construct instruments. In the expert round we added 10 validated instruments, mainly unnamed by their author(s). These instruments were abbreviated with the author names.

We selected the following construct instruments: Aggression: Baron Neuman Geddes Scale (BNGS) [64]; Indirect Aggression Scale—Target (IAS-T) [65]; Fox & Stalworth Scale (FSS) [66]; Cyber aggression, scale [67]; Hospital Aggressive Behavior Scale for coworkers/superiors (HABS-CS) [68]; Healthcare-worker’s Aggressive Behavior Scale-Users (HABS-U) [14].Bullying: Jóhannsdóttir Ólafsson Scale (JOS) [69]; Workplace Bullying Questionnaire—Bullied by Others (WBQ-BO) [70]; Negative Acts Questionnaire: Revised NAQ-R [13]; Cyber NAQ [12]; Escala de Abuso Psicológico Aplicado en el Lugar de Trabajo (EAPA-T) [71]; Patchin & Hinduja Cyber bullying scale (PHCS) [72]; Workplace Cyberbullying Measure (WCM) [73]; Bullying Participants Behavior Questionnaire (BPBQ) [74]; CBQ and CBQ-S (short) [75]; ICA-W [76].Mobbing: Mobbing with effects on the victim [77]; Leidse Mobbing Schaal-ll. (LEMS-II) [78]; (LIPT-60 scale) [79].Harassment/discrimination: Work Harassment Scale (WHS) [80]; Ethnic Harassment Experiences scale (EHE) [81]; Generalized Workplace Harassment Questionnaire (GWHQ) [82]; Cyber harassment [83]; Gender Experience Questionnaire (SEQ) [84]Deviance: Interpersonal and Organizational Deviance Scale (IODS) [31]; Cyber loafing [85,86]; Work related Social Media Questionnaire (WSMQ) [87].CWB: Counterproductive Work Behavior Checklist—long version (CWB-C) [88]Violence: Violence Research health care: in Brazil, Bulgaria, Lebanon, Portugal, South Africa, Thailand, Australia [89].Abuse: Abusive Supervision Scale [90]Terror: Leymann Inventory of Psychological Terror (LIPT) [50]; Cyberstalking [91,92,93].Injustice: Combined effect of perceived organizational injustice and perceived politics on deviant behaviors [94].Interpersonal conflict: Interpersonal Workplace events Scale (IWES) [95]; Interpersonal Conflict at Work Scale (ICAWS) [54].Victimization/scapegoating: Perceived Victimization Scale (PVS) [96]; Juvenile Victimization Questionnaire Interview (JVQ) [97]; Cyberstalking victimization [98].Micropolitics: Perceptions of Organizational Politics Scale (POPS) [99]Ostracism: Workplace Ostracism Scale (WOS) [100]Incivility: Workplace Incivility Scale (WIS) [101]; Uncivil Workplace Behavior Questionnaire (UWBQ) [102]; Cyber incivility [103].Social safety: Social Safety Index at work (SVI) [104]; Monitor of social safety in primary and secondary education [41,105].

This resulted in a total of 44 instruments, of which 39 were validated. The non-validated instruments were: Mobbing with effects on the victim [77]; Violence Research health care in Brazil, Bulgaria, Lebanon, Portugal, South Africa, Thailand, Australia [89]; Cyberstalking [91]; Cyberstalking victimization [98]; Monitoring of social safety in primary and secondary education [41,105]. We still included them because few or no measurement instruments of the constructs in question were found.

However, we also acknowledged that existing measures of NWB might lack aspects that are becoming increasingly important in modern organizations (i.e., cyber-enabled and/or cyber-dependent digital NWB) [106], or that are part of our definition of NWB but not present in existing measures of NWB (i.e., material organizational damage) [107]. Inclusion of this missing information yielded a total of 16 studies on digital NWB, and 32 studies on harm that provided input to the development of the items for the INWBQ.

We selected the following studies on digital NWB: Internet bullying [108]; Cyber-mobbing [109]; Blackmail [110]: Discriminating systems: Gender, race, and power in AI [111]; Cyber fraud [112]; Mail and wire fraud [113]; Technology facilitated violence [114]; Cyber violence on Twitter [40]; Spyware [115]; Problematic Internet Use [116]; Impact of social media on Millennials [117]; Internet politics [118]; Internet ostracism [119]; Cyber incivility [120]; Building digital safety for journalism [121]; Cybercrime, Vandalizing the information society [106].

We selected the following studies on harm: cardiovascular disease and depression [122]; costs [37,107]; psychological stress reactions and physiological stress response [123]; depression of bystanders [124]; sleep difficulties [125]; mental health [21]; job satisfaction and psychological wellbeing [126]; disability retirement [25]; type 2 diabetes [127]; health complaints, social support, personality [128]; psychiatric symptoms [38]; burnout [129]; pain [130]; mental and physical health [131]; job burnout [132]; infectious disease [133]; sickness absence [134]; revanche porn [135]; displaced aggression to family [39]; safety perception [136]; health [137,138]; coronary heart disease [139]; cardiovascular disease [36]; emotional exhaustion in work and family domains [140]; implications for work and family [141]; shame responses [142]; employers & co-workers respond to workplace bullying [143]; employee adiposity [144]; mental health [145].

We listed a total of 44 instruments and 48 studies ordered by construct, including the goal, number of cyber-enabled/-dependent NWB items, questions to actor types and actor roles, number of items on harm, rating scale and anchors, duration, total number of items, and validity. The added instruments in the expert round are indicated with an asterisk; see Appendix A. We further unpack the content of this table in the Section 3.

Several measurement instruments were excluded from our effort to integrate NWB instruments because their items were derived from instruments already included in this study (i.e., Luxembourg Workplace Mobbing Scale (LWMS) [146], Gutenburg Health Study (GHS) [147], Escala de Abuso Psicológico Aplicado en el Lugar de Trabajo Revised (EAPT-T-R), [148], Short Negative Acts Questionnaire (SNAQ) [149]), or because they were not work-related (i.e., Contextual Victimization by Community Violence [150]), or because they addressed coping behaviors rather than NWBs (i.e., De-escalating Aggressive Behavior Scale [151,152]).

### 2.2. Description of Steps in Developing the INWBQ

#### Coding Decisions

From the definition of NWB presented above, we coded the dimensions. To distinguish the NWBs and harm dimensions, we formulated physical harm into bodily harm, material harm into material damage, and psychological harm into mental harm. Because we wanted to develop a questionnaire that is broadly applicable to sectors, we chose the broad group of actor types and coded these dimensions after the model of Merchant and Lundell [15] for stranger, coworker, customer/client/pupil, and relative. The dimensions for actor roles were defined as perpetrator, target, assistant, outsider, and defender, as in the Bullying Participants Behavior Questionnaire (BPBQ) [74] (See Table 1).

We deleted the dimension content that was too general, unclearly worded, or not work-related, combined similar ones, and divided the remaining content into subcategories. We then summed the total number of (highly) similar items in each subcategory. Those NWBs that were included in many instruments (i.e., had the highest number of items) were deemed more important to be included in our instrument and were ranked 1, followed by those with fewer items (ranked 2 or lower). Based on this ranking, we selected the number of items needed to represent each subcategory.

To include the content of two instruments, the Injustice Scale [94] and the Perception of Organizational Politics Scale [99], we reformulated their 24 reverse-scored items into NWBs. This is because these reverse-scored items measure positive rather than negative work behavior.

NWB is a systematically repeating pattern, over a longer period, that can escalate from mild to serious forms [33,81], and be covert without being visible, to overt and visible [153,154], for several months to a year [38,155]. We analyzed the measurement of these patterns (i.e., systematic frequency, duration, escalation, visibility) in the instruments and merged these modes for adoption in our questionnaire.

Since one author (CV) assessed the items in dimensions and subcategories, we examined the inter-rater agreement of all authors in this process. The other authors categorized a random sample of twenty items in dimensions and nine items in subcategories. The inter-rater reliability for both was 100% after discussing some inconsistencies.

Finally, after the coding process was finished, we adjusted the questionnaire on five points (i.e., removing double-barreled parts of selected items, collecting anonymous digital NWB under the covert NWB, asking concrete behaviors, questioning harm, asking mixed participant roles, developing items on escalation and visibility, and adjusting items to external actors). Next, we presented our INWBQ to a panel of experts in this research field. Based on their comments and recommendations, final adjustments were made (see Figure 1).

## 3. Results

### 3.1. Dimensions and Item Selection for NWB, Harm, Actor Types, and Roles

#### 3.1.1. Dimensions of NWB

In Appendix A we list the selected instruments and studies. The abbreviations of the unnamed instruments were taken from the study of Escartín et al. [156]. The goals of the questionnaires could be divided into three types: development (e.g., valid and reliable measure to assess cyberbullying) [73], prevalence (e.g., identify and address workplace violence in the health sector of nine countries) [89], and research on a causal relationship of coping or harm with NWB. Found separate scales for measuring this relation in our studies were the Impact of Event Scale (IES and IES-R (revised) [157], Hospital Anxiety and Depression Scale (HADS) [158], Job Control Scale (QPS) [159], Trauma Evaluation Form (TIF), Posttraumatic Stress Diagnostic Scale (PDS), Maslach Burnout Inventory (MBI) [160], Brief Job Stress Questionnaire(BJSQ) [161], Brief Trauma Questionnaire [162], Violent Incident Form [163] Goldberg scales for anxiety and depression [164], the Demand Control/Support Questionnaire for Stress [165], Occupational Stress Questionnaire (OSQ,) [166], Positive and Negative Affect Scale (PANAS) [167], Outcome Questionnaire (OQ-45.2) [168], Maslach Burnout Inventory (MBI) [169], family undermining survey [39]. Found NWB instruments with scales measuring harm were 11 items of the Turkish version of the LIPT [170], items from the Workplace Sexual Harassment and assault scale [171].

The NWB dimensions were physical, material, psychological, sociocultural, and digital. We filled these dimensions with NWBs from the instrument items and formed subcategories. For instance, for the material NWB dimension, these subcategories were property, resources, job decisions, sabotage, and displays of NWB content. Then, we reduced the number of NWBs further by combining similar ones (e.g., waste and litter). (Due to limited space in this article, full subcategories and their reduction are not included in this article. Here, we suffice with two subcategories of the material NWB dimension and one subcategory of the mental damage dimension to illustrate this process. The full material can be requested from the author). Double-barreled items were distributed separately among the dimensions and subcategories (e.g., causing damage with financial costs was divided among material NWB and material harm). Next, we selected items from the instruments for each final subcategory.

For example, for the dimension material, we selected an item from the NWB “theft” for the subcategory material property, which had the highest rank with a total of 14 highly similar items: “Took items from your desk without prior permission” (UWBQ); and, from the NWB “destruction of property”, the second rank with 12 highly similar items: “Have your personal property or workplace property damaged by someone”(cyber aggression) (see Appendix A, with NWBs of double-barreled and multiple-choice items separately included in red).

In the digital NWB dimension, we made two subcategories reflecting both cyber-enabled NWB and cyber-dependent NWB because they differ. Cyber-enabled NWB concerned traditional NWBs also using a digital medium such as a GSM. Cyber-dependent NWB cannot occur without a digital medium such as identity theft. Cyber, in these two subcategories, is another term for digital, derived from the study of Furnell [106].

#### 3.1.2. Harm Dimensions

The dimensions of harm that we wanted to cover with the measure were bodily harm, material damage, mental harm, and social harm [29]. NWB has been conceived as behavior that harms or threatens to harm others at work (e.g., in workplace aggression [42], bullying [172], workplace harassment [45], counterproductive work behavior [173], and workplace victimization [55]). The NWB instruments ask about several types of inflicted harm due to NWB, such as bodily harm (e.g., physical complaints, SVI) and mental harm (e.g., fear, cyber aggression). On the other hand, various studies used separate questionnaires to show that NWB is associated with bodily harm, i.e., sleep difficulties [125], health impairment [137], cardio-vascular disease [122], and pain conditions [130] (see Appendix A for studies on harm).

These studies also showed that NWB temporarily inflicts mental harm, i.e., psychological distress and stress reactions [123], lower job satisfaction [126], persistent mental complaints, i.e., depression, anxiety, burnout, and PTSD (see review of Verkuil et al. [21]), and turnover intention [174]. Moreover, it has been shown that these outcomes result in material damage, i.e., sickness absence [137], and social harm, i.e., harm to family members [39,97,98] (see Appendix A for studies on harm).

Because four dimensions of harm were part of our integrated NWB definition and some types of harm were not found in the instruments, we included items on physical harm for illness (e.g., diabetes type 2 [127]), material societal damage (e.g., reduced productivity [107]), mental harm in a combination of complaints (e.g., PTSD symptoms, [16,19]), and social harm (e.g., premature retirement [107]) from the studies to fill these dimensions and form subcategories. We reduced the number of harms by removing harms that were too general (e.g., negative psychological outcome), and by combining similar harms (e.g., fear, scared, anxiety). Then we selected items for each subcategory based on the rank number. For example, we selected the item “Did the NWB cause any of the following temporarily mental complaints” with the multiple-choice answers “felt anxious (rank 1), embarrassed (rank 2), stressed out (rank 2), felt angry (rank 3), affect social contacts (rank 4), had difficulties concentrating (rank 5)”. (See Appendix A)

#### 3.1.3. Dimension Actor Types

The subcategories for the dimension actor types were criminal/stranger, customer/client/pupil, coworker/manager, or relative [15]. We gathered the actor types of the instruments for each subcategory, such as Internet user for stranger (cyber fraud). We added a subcategory for actor group of people based on the content of the instruments such as citizens, shared interest group, and organizational group (POPS). After deleting content in subcategories that was too general (e.g., stranger: individual), and combining different names for the same actor types (e.g., worker, subordinate, employee, and colleague were combined into coworker), we obtained our dimension with subcategories for actor types (see Table 2).

Although we found several items to capture actor types, prior to the study we chose to select which actor types are relevant for sampling. Of importance here is to delimit what actor types are considered part of an organization’s network. For instance, this can be done through a realist approach, in which organizations themselves determine the social boundaries, or a nominal approach, in which the boundary is conceptually imposed from the analysis goal [175]. Prior to a survey, a project group can determine the external actors with the help of Table 2.

#### 3.1.4. Dimension Actor Roles

Most of the instruments (*n* = 37) question actors about the role of the target (e.g., being given tasks with unreasonable deadlines, NAQ-R). Other instruments (*n* = 11) question actors as the perpetrator (e.g., acted rudely toward someone at work, Interpersonal and Organizational Deviance scale). A minority (*n* = 5) question bystanders as a witness (e.g., people in this organization attempt to build themselves up by tearing others down, POPS).

The Bullying Participant Behavior Questionnaire (BPBQ) of Demaray [74] questions about three more bystander roles: assistant, defender, and outsider. The assistant is willing to encourage, join in, or aid a bully in bullying others (e.g., “I have made fun of someone when they were pushed, punched, or slapped”). The defender participates in behaviors related to defending on behalf of a victim (e.g., “I tried to make people stop spreading rumors about others”). The outsider acknowledges that bullying occurs but chooses to actively ignore it (e.g., “I pretended not to notice when things were taken or stolen from another student”).

We followed the example of the BPBQ and added the role of witness. In our questionnaire, we combined subcategories referring to bully, aggressor, attacker, perpetrator into perpetrator, and combined subcategories of victim and target into target (see Table 2). In Section 3.1.6 we describe how we adjusted our items to the actor roles.

#### 3.1.5. Occurrence Patterns

The NWB dimension that reflects occurrence patterns was subcategorized as systematic frequency, duration, escalation, and visibility based on the definition of Verschuren et al. [29]. Below we discuss each pattern with our choice for measurement.

##### Systematic

Most NWB instruments (*n* = 23) measure the *systematic* repetitive pattern of NWB as a frequency of occurrence using a 5-point Likert scale. We chose this 5-point scale to leave a center point as an option for a neutral answer [176]. The 3- and 7-point scales also offer this possibility. However, the 3-point scale has too few options, which can make the choice feel forced, while the 7-point scale offers so many options that nuances fall away, and disengagement behavior of respondents increases [177]. We deleted the scales without frequency (e.g., strongly disagree and strongly agree). We combined the 5-point anchors for the INWBQ into the following: *1*—*never*, *2*—*rarely*, *3*—*at least once a month*, *4*—*at least once a week*, *5*—*daily.*

##### Duration

The measurement instruments use different terms for *duration*. We summarized this as *last year at work*. Reporting periods greater than 12 months can lead to potentially serious biases in self-reports due to memory distortions [178,179].

However, when we only ask about the frequency of digital NWB (e.g., weekly), we do not collect information about the weekly amount of use on a medium. By charting the hours of use per day, indicating weekly digital NWB, we have more information on this.

The inventory of place and time of digital use are what Alhabash and colleagues [180] called the viral reach of users. This refers to the interactivity of users on digital media with messages distributed and shared online. We adopted this term for the medium use subcategory with multiple-choice questions about the type of medium (e.g., cloud storage) and duration of medium use (e.g., ½–1 h a day) [108]. We preferred to ask about hours of use, as a percentage usage per day seemed too abstract and difficult to answer.

##### Escalation

As *escalation* is an essential pattern of all NWB labels, it is remarkable that we found only 2 items about escalation: “Not reporting a problem so it would get worse”(CWB-C) [181] and “escalation from online NWB to offline NWBs” (cyberstalking) [91]. Because these two items were quite specific on reporting and online NWB, we developed two general extra items; see Section 3.1.6 under Adjustments.

##### Visibility

It is important to include items on *visibility* because tactics used to remain invisible have a function in disguising the identity of the perpetrator [34,42], and in maximizing harm while minimizing danger to perpetrators, as also indicated with the “effect-danger ratio” [182]. An example of this is the invisible role of the instigator in digital NWB [121]. Another more positive reason for invisible NWB is to keep an option for restoration [183].

However, because most items in the instruments were covert, overt NWB could not be adequately measured. Therefore, following the example in the LIPT-60 by Rivera and Abuín [79], we strived for about one-third of the items to be overt/visible NWB. We also formulated two extra questions on visibility. See Section 3.1.6 for this adjustment.

#### 3.1.6. Adjustments

After establishing dimensions with subcategories for the INWBQ, and selecting items for each of these subcategories, we adjusted some of these items. Our first adjustment involved removing double-barreled parts in the selected items. These are confusing to the respondent and cannot be scored separately. For example, we removed ethnic in the item “Used racial or ethnic slurs to describe you” (FSS).

Another adjustment concerned the anonymous nature of digital NWB. Although this anonymous behavior may be visible, we collect it under the visibility pattern of covert NWB. This is because we consider an unknown perpetrator to be covert NWB.

We decided not to ask about construct names in the items (i.e., “Verbally abused someone at work”, CWB-C). A prior explanation may not preclude interpretations by the respondent. Therefore, we selected concrete and objective behaviors, and we deleted all construct names in our reduction process of items. In this line of reasoning, our third adjustment concerned our way of asking about harm with the construct names as depression, burnout, and PTSD.

These inflicted types of harm by NWB have frequently been examined with separate questionnaires, such as the BDI for depression [184] and IES for PTSD [185]. These questionnaires contain questions on symptoms of dimensions within the relevant constructs. Several NWB questionnaires from this study (e.g., violence research) ask about burn-out, depression, or PTSS inflicted by NWB with one single question. We considered it inappropriate for respondents to self-assess their mental illness. Instead, in three items, we ask them about a combination of symptoms of depression, PTSD, and burnout without naming this a mental illness.

More specifically, for the NWB-inflicted harm depression, we ask about a combination of negative mood [21,98,112], loss of interest (SVI), withdrawal [144], feeling worthless (cyber aggression), feeling embarrassed (GWHQ), and sleep problems (WCM). In doing so, we align with the measurement of depression symptoms [184]. For the NWB-inflicted harm burnout, we ask about a combination of emotional exhaustion [21,114], changed work attitudes (e.g., cynical, less professional) [37,71], low self-esteem (JOS), and alienation (SVI). To make respondents understand this last symptom, we reworded alienation in “not as the one I used to be”. For the NWB-inflicted harm PTSD, we ask about a combination of repeated thinking (SVI), avoiding (FSS), dreams and images of past events (cyber aggression), blame myself (cyber harassment), and emotional reactions as anger (cyber aggression) to align with the measurement of PTSD symptoms [185]. To make respondents understand the question, we reworded repeated thinking as flashbacks, and avoiding as avoid places, people, and activities.

The fourth adjustment concerned asking about mixed actor roles in NWB. To this end, we rewrote all NWB items for the most neutral role of witness (changes in items for this role adjustment were underlined). Escartín et al. [186] showed that different degrees of severity can be questioned, regardless of the participant’s role as a target or witness of NWB. This makes the witness role suitable for interrogating NWB frequencies. In addition, this role provides a good starting point to ask about other roles the respondent fulfilled. For this, we ask the respondent what behavior they performed in the situation in question, with multiple options as a possibility to be ticked; for instance, “someone received unreasonable work demands”, first answer: frequency: 1. never–5. daily, second answer: tick all that apply: I looked away, I encouraged it, I openly disapproved it, etc.

For harm, we also asked the respondent about their role as a witness to the complaints experienced by colleagues and the organization inflicted by NWB. Because this method of questioning could not be applied to the symptoms of depression, PTSD, or burnout, we made the only exception here. For these specific types of harm, we ask whether the respondent experienced these combinations of symptoms themselves.

The fifth adjustment concerned the escalation and visibility occurrence patterns. As we found only two items on escalation in the instruments, we formulated two additional questions for this pattern; for example: “Did the behavior escalate from mild into more serious or physical types over time?” As we found one item on visibility, we formulated two extra items for visibility; for example: “Did the negative behavior change from covert into a more overt and visible nature?” Furthermore, we selected one-third overt items for visibility, that is, a total of 25 overt (indicated in italics) and 42 covert items. Although the covert items are observable by many potential witnesses in the work setting, mental elements such as “trivial”, “purposely”, “unfair”, or “necessary” are not observable, and therefore we considered them as covert NWB.

Our final adjustment concerned the formulation for external actor types. In this way, these questions could be answered by external actor types; for instance, we changed “salary” into “payment”.

#### 3.1.7. Expert Panel

We invited scholars with specific expertise of research on NWB, actor roles, and the development of a questionnaire about actor roles in NWB to take part. We approached eight experts, of which five dropped out: two were out of the office, three had too much work in the planned period, and one of them did not react. Finally, three experts in the field provided answers to our questions.

We asked the expert panel if we were missing important instruments and studies, if they would add additional steps to our method, to consider with us the way of asking about actor roles, and their opinion on the results. Their valuable additions in the literature and studies allowed us to add 10 important instruments (see Appendix A). Some suggested short scales, used in a variety of national cohort studies, but, in the end, these were not included because they contained items from instruments we already included.

Another point of feedback concerned the need to clearly articulate the INWBQ’s practical use in the introduction. How do we want the scale to be used? The purpose of our questionnaire was two-fold: to measure NWB according to the integral definition of NWB and to provide organizations with tools to undertake interventions.

One of the experts pointed out that recreating an essentially longer scale of all NWBs and roles may be a barrier to researchers. If researchers are supposed to pick relevant subscales, why wouldn’t they just pick an established scale such as that in Appendix A? That is of course always possible, but the overlap between instruments weakens the results of such research. In other words, the question remains whether, for example, withdrawal is only a consequence of incivility, as measured with the WIS [144]. When this instrument overlaps with other construct instruments, these results remain obscure. During categorization into subcategories, we indeed encountered a large amount of item overlap between the instruments. Furthermore, to substantiate the item selection by dimension, we ranked the NWBs by summing the total number of items per subcategory. Items that were found among many instruments received the highest rank and were included in the INWBQ. This substantiation can help researchers when considering their choice of an instrument. These two additions also respond to another expert’s request to make a clearer case for why the INWBQ with this breadth, including digital/online behavior, is necessary.

Our steps in the method to develop data, subcategories, design, and items seemed thorough and clearly delineated to the experts. The empirical and theoretical arguments for this method were found to be adequate. Our rationales were both empirical, using this study, and theoretical based on our review and integrated definition.

Furthermore, the experts really challenged us on our way of asking about the different actor roles. For instance, as each subcategory should have items for each role type, this would create a very long list to complete. It was suggested to lessen the number of roles to perpetrator, victim, and bystander, and include within these roles various types of NWB (e.g., physical, verbal). Another expert rightly pointed out that these roles are not fixed but mixed [187]. For instance, research indicated that more than one-half of the bullies also reported being victimized [188]. Because we were determined to stick to our original definition, we formulated the questions in the witness role. Choosing the witness role prevents respondents from viewing themselves in a favorable light, despite not wanting to disclose active involvement as a bully or exaggerating their involvement as a target [189]. Subsequently, for each item, we also ask about which of the five roles the respondent fulfilled. We question these roles regarding concrete behavior per role.

One expert explained to us that there are also “uninvolved” youth in bullying. Therefore, there is not much difference between witness and outsider because some children say they do not see it [188]. As this could also be the case for adults, we chose two very clear behaviors for the outsider role as “I shut my door”, and “I looked away”. Furthermore, it was unclear what instructions would be presented to the person completing the measure. We included an introduction for the INWBQ.

Another important expert addition was about measuring behaviors and consequences in the same questionnaire. In other fields, questionnaires about only consequences have been created independently. As harm is an essential part of the most frequently cited NWB definitions (see our method for these), we included it in the INWBQ. Moreover, combining NWB and its potential harm can help organizations understand the relationship between the risk and its damage, enabling effective risk management, resource allocation, decision making, and compliance. By prioritizing risks based on their potential impact, organizations can allocate resources more efficiently and comply with legal or regulatory requirements [190].

However, properly measuring harm inflicted by NWB has proven to be difficult in past years. Some scholars suggested that NWB could be characterized by the degree of harm it causes. From the target perspective, severity could be measured by the sum of the number of negative actions [191]. This implies that, as the degree of NWB increases, so do the negative outcomes. Other scholars emphasized that it is not the objective behavior but how victims perceive and evaluate this behavior that determines the degree of harm [185,192]. Therefore, Agervold [193] advocated to identify certain types of behaviors, independent of the target perspective.

One application of the above insights is that we inventory harm with a separate Part II, independent of the NWB types identified. This part does not ask about the harm from the target perspective. We ask the respondent to provide an observation of the impact on their environment. This also allows us to inventory the harm for organizations and society that many studies on NWB underline such as sick leave, damaged reputation, and premature retirement.

#### 3.1.8. INWBQ

In Appendix A we present the INWBQ Part I with 69 items on NWBs from existing scales, with in-depth multiple-choice questions about five actor roles and seven items about NWB patterns. Harm is assessed with 11 items (including 50 multiple-choice questions) in Part II of this questionnaire. For the transparency of this development study, in Appendix A we indicate the source of each item and the ranking number of this NWB or harm in the subcategory of its dimension. (See Appendix A).

## 4. Discussion

The purpose of this study was to develop a questionnaire that integrates the content of measurement instruments of 16 most-cited NWB constructs such as bullying, aggression, and discrimination. This was needed since the plurality of different conceptualizations and operationalizations of NWB overlap, threatening their discriminant validity of measurement [194]. In addition, research showed the 83.5% of the different types of NWBs occur simultaneous and sequentially [12], and involve multiple internal and external actor types [15], with more roles than perpetrator and target [195]. Finally, measuring separate NWBs from one source limits our insights into the specific nature of NWB and its inflicted harm to different actor types and roles. It became highly necessary to develop a measurement tool integrating these factors. Our result is the Integrative NWB Questionnaire (INWBQ), which covers five dimensions of NWB in four occurrence patterns, four actor types, six actor roles, and four harms.

### 4.1. Scientific Implications

The vast majority of current NWB measurement instruments focus on physical, psychological, and material NWB, whereas the measurement of sociocultural and digital dimensions in NWB has remained separate from these dimensions. That is, different measurement instruments exist that look at, for example, gender or digital bullying, whereas these NWBs are likely to overlap or co-occur with each other. By recognizing these characteristics in NWB, we contribute to the literature on NWB by illustrating that NWB is a broad construct in which many aspects overlapping with each other. Focusing on only one of these NWB types will create an incomplete picture of reality in an organization.

Another important theoretical contribution concerns our inclusion of four actor types (internal and external to the organization) and six actor roles (target, perpetrator, assistant, defender, outsider, and witness) in addition to the specific type of NWB. Scholars have been urging for several years that data should not only be collected from employees in the target role [26], whereas the majority of information is still collected from this source. The INWBQ identifies external actors in modern workplaces as part of NWB, such as the public, clients, students, and their relatives. These actors are questioned regarding six mixed roles. This will yield new scientific insights into actor types and their roles in NWB. It also illustrates that the influence of actors is not restricted to people internal to the organization, but that increasingly actors outside of the organization influence NWB inside the organization.

Finally, asking about bodily harm, material damage, and mental and social harm for the above-mentioned four actor types and six actor roles is another important scientific implication. It is known that bystanders in the organization develop complaints [196,197]. However, how this varies by actor role and externality provides new insights and research avenues to better address the current challenges that organizations may face. In other words, people other than the target and perpetrator can experience negative consequences based on observing NWB in an organization. The INWBQ allows researchers to also take them into account.

Taken together, this paper advances a much broader picture of NWB that takes into account the overlap between different NWBs, the different actors and their roles involved in NWB, and the harm involved, which together will hopefully facilitate more knowledge that can stimulate interventions to prevent NWB.

### 4.2. Practical Implications

The developed INWBQ consists of two parts. Part I investigates four actor types in six actor roles on physical, material, psychological, sociocultural, and digital NWB. Part II investigates the inflicted bodily harm, material damage, and mental and social harm of these actor types and roles.

For Part I, we recommend to include all modules in organizational research. Individual modules from Part I of the INWBQ do not give a complete picture of an organizational NWB problem, as the probability of combinations of NWB is 83.5% [12]. Taking all modules together will bring these combinations into focus.

If necessary, usage of the separate dimensions of Part I is possible in different research contexts, such as the LWMS [146] for psychological research. For national surveys, such as the GHS [147], separate dimensions are less suitable and further research could develop an abbreviated INWBQ considering the individual types of NWB.

Likewise, we do not recommend using individual modules of Part II of the INWBQ. Porath and Pearson [37] did an excellent job of describing how one type of harm, in time, causes other types. Various health and mental complaints turn into persistent complaints that negatively affect the production process and the reputation of the organization. This also causes material damage through, for example, increased staff turnover and intervention costs.

It is, of course, possible to take Part II separately from Part I when only a picture of the harm and damage is needed and the NWB is known. This can provide additional information for other methods that map harm, such as sick leave analyses, customer complaints, or turnover rates.

Both parts of the INWBQ extract much more information that is valid for organizational consultants, psychologists, and scientists to work with. Part 1 unravels NWBs in different roles and provides new insights to implement specific interventions for bystanders; for example, to increase positive behavior such as defending a target while reducing NWB by assisting a perpetrator. Part II, with the analysis of four types of harm, provides new insights into organizational risks, total costs, and care required. This provides opportunities for system-based interventions involving multiple stakeholders.

### 4.3. Limitations and Future Research

We encountered several limitations in this study. First, in the questionnaires, we could only draw on one example from Demaray [74] on bystander research. This questionnaire did not cover changing roles, as required by our starting point. Moreover, it was not about workers but about youth. Longitudinal research is needed to further investigate changing roles of workers with the INWBQ.

Another limitation of this study is that we could not build on measuring some types of harm from existing NWB instruments (33 Bodily harms, 18 Material damage, 107 Mental harms, 7 Social harms). We added these from additional studies on harm of NWB. As a consequence, we assume, but do not know, whether, for instance, sleep difficulties caused by bullying [125] are also caused by other types of NBW. Integrally including harm in the measurement of various types of NWB, as with the INWBQ, may provide new ways for future research.

A limitation is also that our questionnaire has 76 items in Part I and 11 (50 multiple-choice) items in Part II, creating the risk of non-response. Therefore, attention to procedures to increase the response is necessary. In general, web survey design requires extra attention to procedures to manage the risk of non-response, coverage, and sampling errors [198]. On the other hand, this Internet context provides additional opportunities to make the survey attractive, such as providing context and incremental sampling to reduce dropouts [199]. By grouping the harm items we reduced the time taken to complete the survey, which influences the response rate [200]. Furthermore, we suggest stepwise administration of the questionnaire by sending it in parts to the respondent over a period.

Finally, a limitation was the exclusion of coping because this behavior was not within our baseline definition. Nevertheless, in the NWB instruments we found several items regarding 7 different coping styles: active problem solving (8 items), social support seeking (8 items), avoidance behavior (8 items), palliative reaction (7 items), depressive reaction (11 items), expression of emotions (10 items), and comforting cognitions (1 item) (coping styles after Schreurs et al. [201]). Recent research suggests that these coping styles may mitigate harm, but do not include an effective style for de-escalation of NWB [202]. Therefore, for the inclusion of coping styles within NWB instruments, we believe future research could focus on inclusion of de-escalating styles such as appreciating, reducing anxiety, providing guidance, and maintaining appropriate distance from a person, as measured with the EDABS [151].

Despite these limitations, we believe we have taken a substantial step forward with the INWBQ to provide a complete picture of the facets involved in NWB.

## 5. Conclusions

To the best of our knowledge, this study presents the first attempt to develop an integrated NWB measurement. We succeeded in shaping the content of the INWBQ from 44 NWB measurement instruments and 48 additional studies. With this method we wanted to provide optimal content validity. In this paper we provide a transparent description of the ways we investigated data, built subcategories, and selected and adjusted items.

We developed a full-spectrum diagnostic NWB tool because current tools have shortcomings. They mainly measure one source, i.e., organizational actors in the role of the target. Furthermore, the analyzed instruments show overlapping content, threatening discriminant validity. They do not measure escalation as a key element of all constructs. Meanwhile, new scientific insights for interventions call for measuring different co-occurring types of NWBs, analyzing the dynamics between internal and external actors, and mapping the mixed roles of perpetrator, target, and bystanders.

Although further analytical research on refinement and validation is needed, the INWBQ fills an important research gap relating to these shortcomings. Specifically, Part I allows investigation of four actor types in six actor roles in terms of physical, material, psychological, sociocultural, and digital NWB. In addition, Part II offers insight for organizations about the inflicted bodily harm, material damage, and mental and social harm of these actor types and roles.

## Figures and Tables

**Figure 1 ijerph-20-06564-f001:**
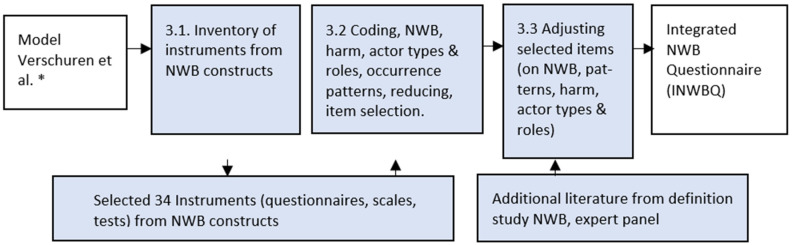
Development INWBQ. * Verschuren et al., 2021 [29].

**Table 1 ijerph-20-06564-t001:** INWBQ dimensions.

Dimensions	1	2	3	4	5	6
NWB	Physical	Material	Psychological	Socio-cultural	Digital	-
Harm	Bodily	Material damage	Mental	Social	-	-
Actor types	Stranger	Worker	Costumer/client/pupil	Relative	-	-
Actor roles	Perpetrator	Target	Assistant	Defender	Outsider	Witness
Occurrence Pattern	Systematic frequency	Duration	Escalation	Visibility	-	-

NWB = negative work behavior.

**Table 2 ijerph-20-06564-t002:** Dimensions Actor Types & Roles, Subcategories.

Dimension Actor Types, Subcategories, Types	Dimension Actor Roles, Subcategories, Roles
**Subcategories stranger** Public, visitors, and users of social media (based on e.g., deviance scale, UWBQ, HABS-U, digital safety) **Subcategory coworker/manager** Co-worker (based on e.g., HABS-CS, cyber aggression, NAQ-R, WCM, CBQ, CBQ-S, ICA-W, LEMS-II, SEQ, EHE, CWB-C, POPS, LIPT, WIS, WOS, UWBQ) Professional e.g., teacher, politician, marketer, journalist, health care worker (based on e.g., HABS-CS, cyberstalking victimization) Manager (based on e.g., UWBQ, WIS, Abusive supervision scale) **Subcategory client/customer/pupil**, Client, customer, students, supplier (based on e.g., cyber incivility, monitor social safety) **Subcategory relative**, Family, acquaintance, (ex) friend, intimate, source (based on e.g., HABS-U, JVQ) **Subcategory actor group** Citizens, shared interest group, organizational group (based on e.g., violence research, POPS)	**Subcategory role perpetrator** Bully, attacker (based e.g., HABS-CS, CWB-C, POPS) **Subcategory role target** Victim, target (based on e.g., ICA-W, EHE, NAQ-R) **Subcategory bystander role witness** See, observe, meaning (based on e.g., Injustice scale, POPS) **Subcategory bystander role assistant** Willingness to encourage, join in, or aid a bully in bullying others (based on BPBQ, SVI) **Subcategory bystander role defender** Participation in behaviors related to defending on behalf of a victim (BPBQ) **Subcategory bystander role outsider** Chooses to actively ignore it (based on BPBQ, SVI, monitor social safety)

## Data Availability

The original contributions presented in the study are included in the article/Appendix A, further inquiries can be directed to the corresponding author/s.

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
