# Peer review of "Beyond Bullying, Aggression, Discrimination, and Social Safety: Development of an Integrated Negative Work Behavior Questionnaire (INWBQ)"

_ijerph, 2023, doi:10.3390/ijerph20166564_

Round 1

Reviewer 1 Report

It is an interesting and valuable article presenting a proposal of Integrated Negative Work Behavior Questionnaire (INWBQ). The authors aim at developing a comprehensive diagnostic instrument allowing for the assessment of Negative Work Behaviors (NWB). In the first stage of the research the authors carried out a thorough literature overview to find (mostly validated) measurement instruments that served as the basis for their concept. In the second stage they discussed the idea with experts. The authors propose adjustments of some items to improve the quality of the questionnaire. The main advantage of the proposal is that INWBQ includes many aspects and is a wide approach to the phenomenon while many other works include a narrow view of the topic. The authors present a multifaceted solution with many dimensions: NWB, harm, actor types, actor roles and occurrence patterns which are described in the literature.

Additionally, I have some particular remarks:

1.      References must be improved as they not follow the guidelines of the journal.

2.      Lines 105-106: something is wrong with tabulation

3.      Line 618 “the probability of 83,5% is much higher that combinations of NWB will occur” is not clear. This sentence must be changed to ensure clarity.

4.      The article contains very long supplementary materials. The authors refer to these materials in the main text but they should underline that the reference is to an external file when it occurs first time in the article, e.g. in lines 228-229: “We listed total 44 instruments and 48 studies in Table 1” but the reader cannot find this table in the text. It becomes clear in line 241 “see Table 1, Supplementary materials” but it is mentioned too late. So I recommend that „supplementary materials” should be given when the Table located in them occurs in the text for the first time.

5.      Table 6 (INWBQ, supplementary materials), p. 1 and p. 6: I would recommend not to use zero as a checkbox, maybe circle or square?

Author Response

1. References must be improved as they not follow the guidelines of the journal.

You are absolutely right about that; we have amended it.

2. Lines 105-106: something is wrong with tabulation

Thank you for this correction. We changed the lines 95-104 as follows: Furthermore, a full measure of NWB should include the different types of harm, namely physical (e.g., cardiovascular disease) [36], material (e.g., replacement costs by employee turnover) [37], psychological (e.g., post-traumatic stress disorder) [38], and social harm (e.g., family consequences) [39]. Additionally, the NWB measure should include the measurement of different actor types involved in NWBs, i.e., strangers/public (e.g., visitor) [14],  workers/managers (e.g., teacher) [40], clients/pupils/customers (e.g., supplier) [41], and relatives (e.g., friend) [14]. Finally, the actor roles need to be assessed which requires the measurement of target, perpetrator, and bystander roles as witness, assistant, defender, outsider. This means that a respondent to the NWB questionnaire can have different roles (e.g., target/perpetrator).

3. Line 618 “the probability of 83,5% is much higher that combinations of NWB will occur” is not clear. This sentence must be changed to ensure clarity.

Thank you for pointing this out. We have now adjusted this as follows in lines 608-611:

For Part I we recommend to include all modules in organizational research. Individual modules from Part I of the INWBQ do not give a complete picture of an organizational NWB problem, as the probability of combinations of NWB is 83.5% [12]. Taking all modules together will bring these combinations into focus.

4. The article contains very long supplementary materials.

The authors refer to these materials in the main text, but they should underline that the reference is to an external file when it occurs first time in the article, e.g. in lines 228-229: “We listed total 44 instruments and 48 studies in Table 1” but the reader cannot find this table in the text. It becomes clear in line 241 “see Table 1, Supplementary materials” but it is mentioned too late. So I recommend that „supplementary materials” should be given when the Table located in them occurs in the text for the first time.

Thank you very much for pointing this out to us, we overlooked this point and have amended it in line 225-226 as follows:

See Table 1, Supplementary materials.  We will further unpack the content of this table in the Results section.

5. Table 6 (INWBQ, supplementary materials), p. 1 and p. 6: I would recommend not to use zero as a checkbox, maybe circle or square?

Thank you very much for this suggestion, we made squares for the checkbox.

Reviewer 2 Report

The subject of the work is interesting.

The following comments and recommendations are made:

The article has no theoretical framework. It is necessary to establish a theoretical framework for the development of the research.

The introduction is somewhat lengthy. 

At the end, it would be useful to indicate the structure of the paper.

2.1. Selection of instruments

The selection of instruments should be better explained, making it clearer which instruments have not been validated.

3.3.7. Expert panel

With regard to the panel of experts, it is not well justified why three and not more experts. There is no doubt about the capacity of the experts, but there is also no indication of the requirements that the experts had to have in order to become experts. This part should be improved.

4. Discussion

In the Discussion section, it should be made clearer what the added value of the work is both on a theoretical and practical level. Thus, for example, for the professional level, what the integrative questionnaire entails.

4.3 Limitations

The limitations of the study need to be expanded and better explained. 

5. Conclusions

There is room for improvement in the Conclusions section. 

Author Response

1.The article has no theoretical framework. It is necessary to establish a theoretical framework for the development of the research.

We did not mention it a theory but a specified a new integrative definition of NWB in lines 75-85:

This developmental breadth is searched in the coverage of NWB types with co-occurrence patterns, types of harm, actor types and actor roles in accordance with the model of Verschuren et al. [29]. These authors reviewed the NWB field and identified the overlapping and unique aspects of the operationalizations of NWBs to specify a new integrative definition of NWB. The key elements in their definition are: 1) the idea that NWBs are distinguished in five natures (i.e., physical, material, psychological, sociocultural, and/or digital); 2) are associated with different types of harm (i.e., physical, psychological, material, and social); 3) are engaged in by different actors (i.e., criminal/stranger, customer/client/pupil, coworker/manager, or relative); 4) can be performed by individuals, dyads, triads or groups; and 5) are often taking place over time.

2.The introduction is somewhat lengthy. 

At the end, it would be useful to indicate the structure of the paper.

Although this point is not indicated in the template of Ijerph we agree with you that the introduction of 1.118 words is quite lengthy. This is due to our choice of including the extended definition that is our starting point.

We added the structure of the paper in lines 112-117 as follows:

This article is organized as follows. First, we explain our development method for the INWBQ on dimensions and item selection. Then we present our results for the dimensions of NWB, occurrence patterns, harm, actor types and actor roles. Next, we explain which adjustments were made and why. Thereafter, we indicate how we processed the contributions from the expert panel. We end with a discussion, scientific and practical implications, limitations, and future research.

2.1. Selection of instruments

The selection of instruments should be better explained, making it clearer which instruments have not been validated.

Thank you very much for this suggestion. We adjusted the explanation as follows in lines 208-215:

This search yielded a total of 34 construct instruments. In the expert round we added 10, mainly unnamed by its author(s), validated instruments. This resulted in a total of 44 instruments of which 38 were validated. The not validated instruments were: Mobbing with effects on the victim [64], Violence Research health care: in Brazil, Bulgaria, Lebanon, Portugal, South Africa, Thailand, Australia [65], Cyberstalking [66], Cyberstalking victimization [67], Monitor of social safety in primary and secondary education [41, 68], Building digital safety for journalism. [69]. We still included them because few or no measurement instruments of the constructs in question were found.

3.3.7. Expert panel

With regard to the panel of experts, it is not well justified why three and not more experts. There is no doubt about the capacity of the experts, but there is also no indication of the requirements that the experts had to have in order to become experts. This part should be improved.

Thank you for drawing our attention to this important point. We certainly like to explain in more detail on which qualifications we selected the experts.  It is also right that you point out that we do not justify why there were three experts and not more. In lines 480-484 we formulated this as follows:

We invited scholars with specific expertise of research on NWB, actor roles, and the development of a questionnaire on actor roles in NWB. We approached eight experts of which five dropped out: two were out of office, three had too much work in the planned period, and one of them did not react. Finally, three experts in the field, provided answers to our questions.

4. Discussion

In the Discussion section, it should be made clearer what the added value of the work is both on a theoretical and practical level. Thus, for example, for the professional level, what the integrative questionnaire entails.

Very good that you point out to us that the Discussion section can be made clearer on theoretical and practical level. Thank you. We have now adjusted it in this way in lines 575-634:

4.1. Scientific implications

   The vast majority of current NWB measurement instruments focus on physical, psychological, and material NWB, whereas the measurement of sociocultural and digital dimensions in NWB has remained separate from these dimensions. That is, different measurement instruments exist that look at for example, gender or digital bullying whereas these NWBs are likely to overlap or co-occur with each other. By recognizing these characteristics in NWB, we contribute to the literature on NWB by illustrating that NWB is a broad construct with many aspects overlapping with each other. Focusing on only one of these NWB types will create an incomplete picture of reality in an organization.

      Another important theoretical contribution concerns our inclusion of four actor types (intern and extern to the organization) and six actor roles (target, perpetrator, assistant, defender, outsider, and witness) in addition to the specific type of NWB. Scholars have been urging for several years that data should not only be collected from employees in the target role [26], whereas still the majority of information is collected from this source. The INWBQ identifies external actors in modern workplaces as part of NWB, such as the public, clients, students,and their relatives. These actors are questioned on six mixed roles. This will yield new scientific insights on actor types and their roles in NWB and illustrates that the influence of actors is not restricted to people internal to the organization but that increasingly actors outside of the organization influence NWB inside the organization.

      Finally, asking about bodily harm, material damage, mental and social harm on the above mentioned four actor types and six actor roles is another important scientific implication. It is known that bystanders in the organization develop complaints [126, 127]. However, how this varies by actor role and externality provides new insights and research avenues to better address the current challenges that organizations may face. In other words, other people than the target and perpetrator can experience negative consequences based on observing NWB in an organization. The INWBQ allows researchers to take them into account as well.

    Taken together, this paper advances a much broader picture of NWB that takes into account the overlap between different NWBs, the different actors and their roles involved in NWB, and the harm involved that together will hopefully facilitate more knowledge that can stimulate interventions to prevent NWB.

4.2. Practical implications

    The developed INWBQ consists of two parts. Part I investigates four actor types in six actor roles on physical, material, psychological, sociocultural, and digital NWB. Part II investigates the inflicted bodily harm, material damage, mental and social harm of these actor types and roles.

     For Part I we recommend to include all modules in organizational research. Individual modules from Part I of the INWBQ do not give a complete picture of an organizational NWB problem, as the probability of combinations of NWB is 83.5% [12]. Taking all modules together will bring these combinations into focus.

     If necessary, usage of the separate dimensions of Part I is possible in different research contexts, such as the LWMS [72]for psychological research. For national surveys, such as the GHS [73] separate dimensions are less suitable and further research could develop an abbreviated INWBQ considering the individual types of NWB.

     Likewise, we do not recommend using individual modules of Part II of the INWBQ. Porath and Pearson [37] did an excellent job of describing how one type of harm in time causes other types. Various health and mental complaints turn into persistent complaints that negatively affect the production process and the reputation of the organization. It also causes material damage through, for example, increased staff turnover and intervention costs.

     It is of course possible to take Part II separately from Part I when only a picture of the harm and damage is needed and the NWB is known. This can provide additional information for other methods that map harm, such as sick leave analyses, customer complaints or turnover rates.

     Both parts of the INWBQ bring out much more and valid information for organizational consultants, psychologists, and scientists to work with. Part 1 unravels NWBs in different roles and brings new insights to implement specific interventions for bystanders. For example, to increase positive behavior such as defending a target while reducing NWB by assisting a perpetrator. Part II, with the analysis of four types of harm, brings new insights into organizational risks, total costs and care required. This provides opportunities for system-based interventions involving multiple stakeholders.

4.3 Limitations

The limitations of the study need to be expanded and better explained. 

Thank you for this suggestion. We agree that the limitations could be described in more detail. Elsewhere in the discussion, we provide limitations that fit better under the heading limitations and future research. We have changed this section in lines 636 -668 as follows:

4.3. Limitations and future research

     We encountered several limitations in this study. First, in the questionnaires, we could only draw on one example from Demaray [78] on bystander research. This questionnaire did not cover changing roles as our starting point required. Moreover, it was not about workers but about youth. Longitudinal research is needed to further investigate changing roles of workers with the INWBQ.

     Another limitation of this study is that we could not build on measuring some types of harm from existing NWB instruments[1]. We have added these from additional studies on harm of NWB. As a consequence, we assume but do not know whether for instance sleep difficulties caused by bullying [90] are also caused by other types of NBW. By integrally including harm in the measurement of various types of NWB, as with the INWBQ, this may provide new ways for future research.

      A limitation is also that our questionnaire has 76 items in Part I and 11 (50 Multiple choice) items in Part II, causing the risk of non-response. Therefore, attention to response increasing procedures is necessary. In general, web survey design requires extra attention to procedures to manage the risk of non-response, coverage, and sampling errors [128]. On the other hand, this Internet context provides additional opportunities to make the survey attractive, such as providing context and incremental sampling to reduce dropouts [129]. By grouping the harm-items we reduced the time taken to complete the survey influencing the response rate [130]. Furthermore, we suggest stepwise administration of the questionnaire by sending it in parts to the respondent over a period.

     Finally, a limitation was the exclusion of coping because this behavior was not within our baseline definition. Nevertheless, in the NWB instruments we found several items of seven different coping styles: active problem solving (8 items), social support seeking (8 items), avoidance behavior (8 items), palliative reaction (7 items), depressive reaction (11 items), expression of emotions (10 items), and comforting cognitions (1 items) (coping styles after Schreurs et al. [131]). Recent research suggests that these coping styles may mitigate harm, but do not include an effective style for de-escalation of NWB [132]. Therefore, for the inclusion of coping

styles within NWB instruments, we believe future research could focus on inclusion of de-escalating styles such as appreciating, reducing anxiety, providing guidance, maintaining appropriate distance from a person, as measured with the EDABS [77].

    Despite these limitations, we believe we have taken a substantial step forward with the INWBQ to provide a complete picture of the facets involved in NWB.

5. Conclusions

There is room for improvement in the Conclusions section. 

We agree that the conclusion may include more elements from our goal and results. We have adapted it in lines 671-688:

    To our knowledge, this study presents the first attempt to develop an integrated NWB measurement. We succeeded in shaping the content of the INWBQ from 44 NWB measurement instruments and 48 additional studies. With this method we wanted to provide optimal content validity. In this paper we gave a transparent description of the ways we tapped data, built subcategories, selected, and adjusted items.

    We developed a full-spectrum diagnostic NWB tool because current tools have shortcomings. They mainly measure one source, i.e., organizational actors in the role of target. Furthermore, the analyzed instruments show overlapping content, threatening discriminant validity. They do not measure escalation as a key element of all constructs. Meanwhile, new scientific insights for interventions call for measuring different co-occurring types of NWBs, analyzing the dynamics between internal and external actors, and mapping the mixed roles of perpetrator, target, and bystanders.

    Although further analytical research on refinement and validation is needed, the INWBQ fills an important research gap on these shortcomings. Namely, Part I allows investigation of four actor types in six actor roles on physical, material, psychological, sociocultural, and digital NWB. In addition, part II offers insight for organizations on the inflicted bodily harm, material damage, mental and social harm of these actor types and roles.

[1] 33 Bodily harms, 18 Material damage, 107 Mental harms, 7 Social harms.

Reviewer 3 Report

This article is innovative and original with a great interest in the area.

Small recommendations are described below:

1. About the REFERENCES in the text: Authors need to follow the reference numbers should be placed in square brackets [ ], and placed before the punctuation; for example [1], [1–3], or [1,3]. For embedded citations in the text with pagination, use both parentheses and brackets to indicate the reference number and page numbers; for example [5] (p. 10). or [6] (pp. 101–105).

2. In section 2.1 Selection of Instruments - it is not clear how they arrived at these instruments - which databases they consulted with the 16 most frequently cited NWB constructs as keywords, where? or followed Verschuren et al. (2021)? It is not clear what the decision was made.

3. Pay attention to gender equality issues - instead of "his" or "her" - put "their".

4. We suggest reviewing the entire article and standardizing the punctuation: Examples: lines 379, 398: Verschuren et al., (2021) - the comma, in this case, is to remove.

5. Line 572: (Privitera and Campbell, 2009), should be (Privitera & Campbell, 2009).

6. I suggest reducing the font size of the content of some tables to be proportional to the text.

Apart from these recommendations, the article is very good and suitable for publication in this journal.

Author Response

1. About the REFERENCES in the text: Authors need to follow the reference numbers should be placed in square brackets [ ], and placed before the punctuation; for example [1], [1–3], or [1,3]. For embedded citations in the text with pagination, use both parentheses and brackets to indicate the reference number and page numbers; for example [5] (p. 10). or [6] (pp. 101–105).

Very thoughtful of you to point this out to us with these additional clarifications. We have applied these. Thank you very much.

2. In section 2.1 Selection of Instruments - it is not clear how they arrived at these instruments - which databases they consulted with the 16 most frequently cited NWB constructs as keywords, where? or followed Verschuren et al. (2021)? It is not clear what the decision was made.

Thank you for bringing this to our attention. We used the literature with references from the 2021 review to search tools. We added this information as follows in lines 127-128:

      We searched for these instruments in the review of Verschuren et al.[29] that identified the 16 most frequently cited NWB constructs as keywords [29]:

3. Pay attention to gender equality issues - instead of "his" or "her" - put "their".

Thank you for this suggestion. Table 6 is adjusted on this point.

4. We suggest reviewing the entire article and standardizing the punctuation: Examples: lines 379, 398: Verschuren et al., (2021) - the comma, in this case, is to remove.

Thank you for this suggestion, we improved this.

5. Line 572: (Privitera and Campbell, 2009), should be (Privitera & Campbell, 2009).

Thank you, with the other citation style this point is automatically resolved.

6. I suggest reducing the font size of the content of some tables to be proportional to the text.

Apart from these recommendations, the article is very good and suitable for publication in this journal.

We can imagine this would enhance readability and will ask the publisher if this is possible.